# From movement to METs: A validation of ActTrust® for energy expenditure estimation and physical activity classification in young adults

Elias dos Santos Batista[1,2,3], Stephania Ruth Basilio Silva Gomes[4],
Ayrton Bruno de Morais Ferreira[1], Lucas G. S. França[5,6], John Fontenele Araújo[3,4,7],
Arnaldo Luis Mortatti[1], Mario André Leocadio-Miguel[8]*

**1** Programa de Pós-Graduação em Educação Física, Universidade Federal do Rio Grande do Norte, Natal, Rio Grande do Norte, Brazil, **2** Instituto Federal de Educação Ciência e Tecnologia do Rio Grande do Norte, Caicó, Rio Grande do Norte, Brazil, **3** Programa de Pós Graduação Multicêntrico em Ciências Fisiológicas, Universidade Federal do Rio Grande do Norte, Natal, Rio Grande do Norte, Brazil, **4** Programa de Pós-Graduação em Psicobiologia, Universidade Federal do Rio Grande do Norte, Natal, Rio Grande do Norte, Brazil, **5** School of Computer Science, Faculty of Science and Environment, Northumbria University, Newcastle upon Tyne, United Kingdom, **6** Department of Forensic and Neurodevelopmental Science, Institute of Psychiatry, Psychology & Neuroscience, London, United Kingdom, **7** Laboratório de Neurociência e Psicologia Social - LaNPSo, Universidade Federal do Delta do Parnaíba, Parnaíba, Piauí, Brazil, **8** School of Psychology, Faculty of Health and Wellbeing, Northumbria University, Newcastle upon Tyne, United Kingdom

* mario.miguel@northumbria.ac.uk

## Abstract

Estimating physical activity (PA) levels is a challenging and expensive task. An alternative could be the use of actigraphy devices to estimate PA. This has been previously done to a number of devices, including ActiGraph® GT3X+. In this study, we validated ActTrust® against the widely used GT3X+ and compared activity counts to metabolic equivalents (METs) derived from indirect calorimetry during treadmill walking and running. Fifty-six young adults (34 men, 22 women) participated in controlled effort exercises including light, moderate, vigorous, and very vigorous activity intensities. We developed a linear model to estimate energy expenditure (EE) from movement count of combinations of devices placed at hip or wrist. We then estimated cut-off points for each intensity range. Our results showed correlations between treadmill speed and both METs ($r=0.95$, $p<0.05$) and movement counts from both GT3X+ and ActTrust devices placed either on the hip ($r=0.94$, $p<0.05$; $r=0.93$, $p<0.05$) or on the wrist ($r=0.88$, $p<0.05$; $r=0.88$, $p<0.05$), respectively. Our proposed model performed well with balanced accuracies above 0.77 for all intensity ranges and over 0.9 for light and moderate activity. This is the first study to model estimate and validate PA intensity thresholds on ActTrust® devices. Our findings support the use of ActTrust® devices as simple, cost-effective tool for 24-hour assessments of EE.

**Data availability statement:** The data and source code used in the analyses featured in this article are available at https://github.com/circadia-bio/ACTT_validation_study.

**Funding:** The author(s) received no specific funding for this work.

**Competing interests:** The authors have declared that no competing interests exist.

## Introduction

Physical activity (PA) is recognised for providing several health benefits in humans, mainly preventing and controlling chronic non-communicable diseases [1]. Furthermore, the practice of PA is associated with improved subjective and mental well-being and physical self-perception [2]. The World Health Organization (WHO) recommends 150–300 minutes of moderate-intensity aerobic PA, or at least 75–150 minutes of vigorous-intensity aerobic PA, or an equivalent combination of PA of both intensities during the week for adults. Also important is the WHO's global action plan on physical activity, which aims to decrease the prevalence of sedentarism in adults and adolescents by 15 per cent by 2030 [3].

One way to define the intensity of PA is through the metabolic equivalent (MET) of task. This mechanism expresses energy expenditure (EE) and indicates the oxygen needs for different activities, which at rest is equivalent to 3.5 ml $O_2$/ kg/min [4,5]. The gold standard methodological approach to assessing energy expenditure is the doubly labelled water (DLW) technique [6]. However, there are limitations mainly related to this method's high cost, complexity and scalability. Alternatively, accelerometry emerges as a practical, scalable and economically viable method to estimate energy expenditure (EE) and quantify PA based on movement and body acceleration, generating numerical variables of frequency, duration and intensity of the actions performed [7,8]. The challenges inherent in using accelerometry include the diversity of designs in accelerometer models, the disparities in signal processing between the different sensors and the multiplicity of options regarding possible placements [9–11]. Furthermore, it is possible to verify a variation in the algorithms that calculate the counts, both of the activity and other extracted parameters on different devices [12].

The ActiGraph® GT3X+ accelerometer has the most significant number of validation studies and published thresholds and has been widely used to validate devices with the same purpose [13]. The ActTrust® (Condor Instruments, São Paulo, Brazil) is a device that has a triaxial accelerometer and is specific for inferring sleep from locomotor activity [14]. It offers several advantages over other actigraphy devices: (1) significantly lower cost, making it accessible for large-scale studies in resource-limited settings; (2) longer battery life enabling extended monitoring periods; (3) integrated light and temperature sensors for comprehensive physiological assessment. However, despite these practical advantages, the ActTrust® has not been validated against gold-standard measures of EE, which limits its use in research and clinical contexts.

The validation of accelerometers through indirect calorimetry is crucial for developing mathematical equations capable of predicting energy expenditure using these devices and establishing cutoff points for classifying the intensity of PA derived from accelerometer data [15]. Based on this context, the two main objectives of this study are: (i) to define equations for estimating EE using ActTrust in young adults, and (ii) to establish cutoff points for the ActTrust movement variable (light, moderate, vigorous and very vigorous PA).

## Materials and methods

### Participants

Fifty-six healthy young adults were recruited for this study by online advertising. We established musculoskeletal or cardiovascular diseases that could hinder PA as excluding criteria. Participants were also excluded if they had any other contraindications to exercise or were taking medication that affected their metabolic rate. All participants completed the Physical Activity Readiness Questionnaire (PAR-Q) [16–18], and no one was excluded from the study for answering yes to one or more questions. The statistics of the characteristics of the participants are presented in Table 1.

The study was approved by the Human Ethics Committee of the Federal University of Rio Grande do Norte, under protocol number 3.672.214 (CAAE: 15857419.5.0000.5537). It was performed according to the declaration of Helsinki and complied with the Ethical Standards in Sport and Exercise Science Research. Participants provided written consent to participate in the study. Recruitment took place between March 14, 2022 and October 24, 2022.

The sample size of 56 participants used in this study is consistent with previous validation research comparing accelerometer-based activity monitors against indirect calorimetry. Similar studies validating energy expenditure estimates from devices such as the ActiGraph GT3X+ have used samples ranging from 31 to 50 participants [8,19]. Given the similarity in the study design, controlled laboratory setting and repeated measures across multiple activity intensities and device placements, our sample size provides sufficient statistical power to detect meaningful associations and develop robust predictive models.

### Measuring activity

The *ActiGraph GT3X+* (ActiGraph Corporation, headquartered in Pensacola, FL, Firmware version 4.1.0) is a lightweight device that employs triaxial accelerometers and offers a dynamic range ranging from −6 g to +6 g. For the current study, a sampling rate of 30 Hz was chosen, as this range is expected to capture the majority of accelerations arising from human movement sufficiently [20]. After digitisation, the signal passed through a digital filter that restricts the accelerometer's frequency range to 0.25–2.5 Hz. We used the vector magnitude (VM) data (vector representing the summed value of the three axes). Data from the GT3X+ device were retrieved and processed using *ActiLife version 6.13.4*, software provided by ActiGraph Corporation based in Pensacola, FL.

The ActTrust® device (Condor Instruments, São Paulo, SP, Brazil) is an actigraphy tool that features a rechargeable battery, a digital tri-axial accelerometer, a skin temperature sensor and a light sensor. It samples acceleration data at 25 Hz, ranging from 0.03 to 4 g. First, this data undergoes processing through a band-pass filter (0.5–2.7 Hz), after which the norm is calculated from the three axes. The resultant value is then integrated over an epoch (1–60 seconds) and stored in memory. This equipment is designed for activity and sleep monitoring and processes activity using parameters such as PIM (proportional integration mode), ZCM (zero crossing mode) and TAT (time above threshold) [21]. The *Act Studio software* (Condor Instruments, São Paulo, Brazil) downloads, visualises, and exports the collected data.

Importantly, this study used the highest temporal resolution available for both actigraphy devices, according to their specific firmware. Each 1-second epoch was calculated by aggregating filtered and rectified raw acceleration data into

**Table 1. Demographic and physical characteristics of the participants. Mean (st. dev.).**

| Sex | Sample size | Age (y) | Height (cm) | Weight (kg) | Body mass index (kg/m²) |
|---|---|---|---|---|---|
| Male | 34 | 28.3 (4.6) | 172 (6.2) | 78.2 (11.9) | 23.1 (1.6) |
| Female | 22 | 26.8 (5.0) | 160 (7.3) | 59.9 (9.4) | 26.7 (2.5) |

a single 1-second bin. This was consistent with validation approaches for wrist and hip-worn accelerometers across the same range of walking speeds [8,19].

## Measuring oxygen uptake

Oxygen uptake was measured 'breath-by-breath' continuously during each condition using indirect calorimetry (*Quark CPET*, Cosmed®, Italy). Before testing each subject, we calibrated the metabolic cart with a known gas mixture (16% $O_2$ and 5% $CO_2$) and volume. One test had to be repeated seven days later due to an error in the treadmill's security system (*Super ALT*, Inbramed®, Porto Alegre, Brazil). Occasional errant breath episodes (e.g., due to coughing, swallowing or talking) were excluded from the data set when exceeding three standard deviations of the mean, the latter being defined as the average of two following and two preceding sampling intervals [22].

## Experimental procedure

The entire data collection took place between March 21, 2022, and November 1, 2022. All *GT3X+* and *ActTrust* devices were initialiased via computer interfaces to collect data in 1-second epochs [19]. The GT3X + was initialised to collect the VM data, and the ActTrust to collect activity in the PIM mode. Each participant wore two pairs of GT3X+ and ActTrust devices simultaneously, securely placed on the hip on the side of the participant's dominant lower limb through an elastic belt and the other on the non-dominant wrist.

Participants refrained from exercise and fasted for at least 4 hours before the testing session. The protocol consisted of 5 conditions (of 10-min duration each), using a progressive incremental design, interspersed with 5-minute rest periods: (i) resting; (ii) treadmill (Quasar Med 4.0, h p cosmos, Nussdorf-Traunstein, Germany) walking at 3 $km \cdot h^{-1}$; (iii) treadmill walking at 5 $km \cdot h^{-1}$; (iv) treadmill walking or running at 7 $km \cdot h^{-1}$; and (v) treadmill running at 9 $km \cdot h^{-1}$. Prior to initiating the protocol, devices were synchronised using a manual time-matching approach [23], allowing precise temporal alignment between the accelerometers and breath-by-breath metabolic data.

## Data analysis

To ensure that data reflected true physiological steady states, we adopted the protocol described in previous studies [8,19]. Thus, we calculated the mean activity counts from the four central minutes of the Vector Magnitude (VM) for GT3X+ and the counts (proportional integral mode – PIM) for *ActTrust* to estimate activity levels. METs (Metabolic Equivalents) were individually calculated (VO²) similarly.

First, we assessed the effect of the different devices and placement with an ANOVA according to the model Treadmill speed$\sim$ Activity$^{1/2}$ : Device$_{placement}$. Secondly, we then modelled the association between METs and activity for each device using a general linear model defined in equation 1. A square root transformation was applied to both METs and activity counts due to the non-linear relationship. Model assumptions were verified through examination of residual plots and Q-Q plots for normality – included in supplementary material S3 Fig in S1 File.

$$MET^{1/2} \sim Activity^{1/2} * Device_{placement} \tag{1}$$

We evaluated two devices according to their placement, hence our model can be written as in equation 2.

$$MET^{1/2} = \beta_0^{\dagger} + \beta_1^{\dagger} Activity^{1/2} \tag{2}$$

with $\beta_0^{\dagger} = \beta_0 + \beta_2 ACTT_{hip} + \beta_3 ACTT_{wrist} + \beta_4 GT3X+_{wrist}$ and $\beta_1^{\dagger} = \beta_1 + \beta_5 ACTT_{hip} + \beta_6 ACTT_{wrist} + \beta_7 GT3X+_{wrist}$. $ACTT_{hip}$, $ACTT_{wrist}$, and $GT3X+_{wrist}$ are binary variables set to adjust the model values to each configuration. $GT3X+_{hip}$ was defined as the reference level.

Analyses were conducted using R v. 4.4.1 (R Foundation for Statistical Computing, Vienna, Austria) [24] and auxiliary packages *tidyverse* v. 2.0.0 [25], *caret* v. 6.0.94 [26], *MuMIn* v. 1.48.4 [27], *MetBrewer* v. 0.2.0 [28], *cowplot* v. 1.1.3 [29], *pROC* v. 1.18.5 [30], *broom* v. 1.0.6 [31], *lmtest* v. 0.9.40 [32], *car* v. 3.1.2 [33], *effectSize* v. 1.0.2 [34], and *msm* v. 1.8.2 [35]. The significance level was set at $\alpha = 0.05$. The data and scripts used in the analyses featured in this article are available at https://github.com/circadia-bio/ACTT_validation_study [36].

We estimated metabolic rate in METs to derive the cut-points for moderate, vigorous and very vigorous exercise intensity levels according to the intervals (3.0–5.99, 6.0–8.99 and above 9.0 METs).

We employed performance metrics for model diagnostics and computed $R^2$ and confusion matrix. Finally, we evaluated the ability of the new cut points from the linear regression model to accurately classify the physical activity intensity levels derived from EE objective measurements, according to the different MET bands with sensitivity, specificity, balanced accuracy, and area-under-the-curve (AUC) bench marks in a similar fashion to previous works in literature [8].

## Results

The sample comprised young adults (34 men, 22 women) aged 18–35. The participants' characteristics are displayed in Table 1. Energy expenditure values obtained with indirect calorimetry reflected each condition, increasing proportionally according to the treadmill speed, see Fig 1.

### Activity counts and treadmill velocity

We performed a two-way ANOVA to understand the outputs of each device (GT3X+ and ActTrust) according to the placement and condition (treadmill velocity). Movement counts from the GT3X+ devices were mostly distinct (F(19, 1087) = 1347, $p < 0.001$, $\eta^2 = 0.96$, 95% CI [0.96, 1.00]) with significantly lower activity magnitudes than those from the ActTrust devices at 3, 5, 7 and 9 $km \cdot h^{-1}$ ($p < 0.05$).

In contrast, placement did not affect the measures for GT3X+ regardless of hip or wrist acquisitions for 3 $km \cdot h^{-1}$ and 5 $km \cdot h^{-1}$. Movement counts of both devices (GT3X+ and ActTrust) placed on the hip or wrist increased with the increment of the treadmill speed (see Fig 1). Statistical details of the ANOVA analysis and post-hoc Tukey's Honestly Significant Difference (HSD) [37,38] tests were included in the supplementary table S1 in S1 File.

### Definition of equations to estimate energy expenditure

Our main objective was to analyse the relationship between motion measurements obtained with the ActTrust® in comparison to the GT3X+® and metabolic equivalent based on gas exchange in young adults. First, we explored how treadmill speed is associated with METs and movement counts. Specifically, speed was highly correlated to both METs ($r = 0.95$, $p < 0.05$) and movement counts from both GT3X+ and ActTrust devices placed either on the hip ($r = 0.94$, $p < 0.05$; $r = 0.93$, $p < 0.05$) or on the wrist ($r = 0.88$, $p < 0.05$; $r = 0.88$, $p < 0.05$), respectively. Our second step was to develop a linear regression model to estimate METs based on the movement counts recorded by each device at each placement (see Fig 2c). The resulting equations are crucial for estimating physical activity in METs from the movement counts of the GT3X+ and ActTrust devices placed on the hip and wrist (see Table 2).

### GT3X+® and ActTrust® movement count cut points to classify physical activity intensity

Finally, we derived cut-points based on MET ranges [0,3) or light, [3,6) or moderate, [6,9) or vigorous, [9,∞) or very vigorous physical activity intensity ranges based on the movement counts from both devices and placement. We then assessed the ability of the new cut points from our linear regression model to accurately classify the physical activity intensity levels objectively derived from EE measurement (METs). We estimated the sensitivity (recall), specificity, balanced accuracy, area-under-the-curve (AUC), and confusion matrix for each intensity zone (for metrics see Table 3). The

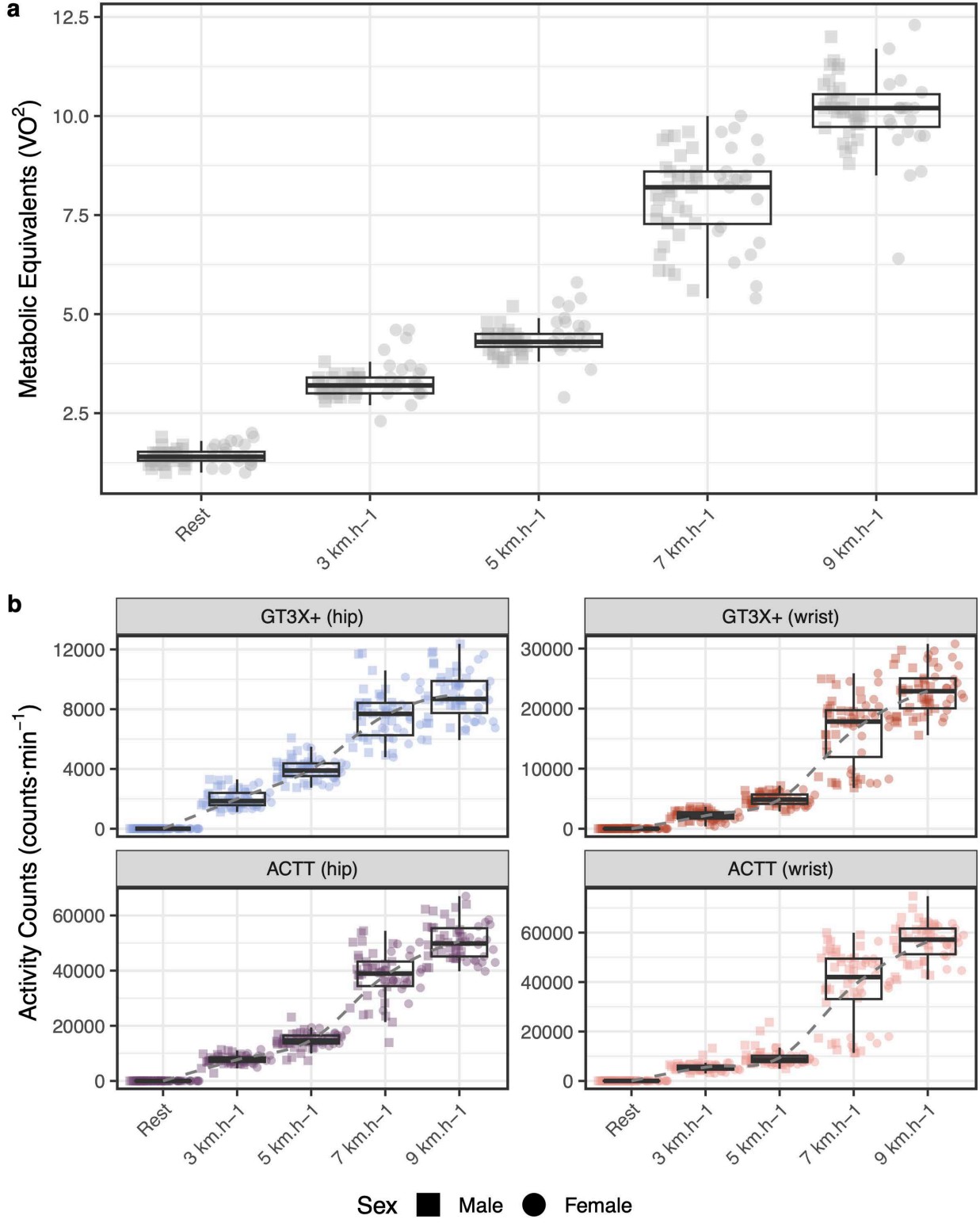

**Fig 1. Scores for each treadmill speed band. a)** Metabolic equivalents ($VO^2$) obtained from all participants for each speed band. **b)** Activity counts obtained for each speed band for every combination of placement and device, i.e., *GT3X+ (hip), ACTT (hip), ACTT (wrist), GT3X+ (wrist)*. The scores across the different devices show a similar profiles increasing towards higher speeds, which is compatible to the variation exhibited by the ergo-spirometry metabolic equivalents measures. Point shape denotes sex (square: male, $n = 34$; circle: female, $n = 22$).

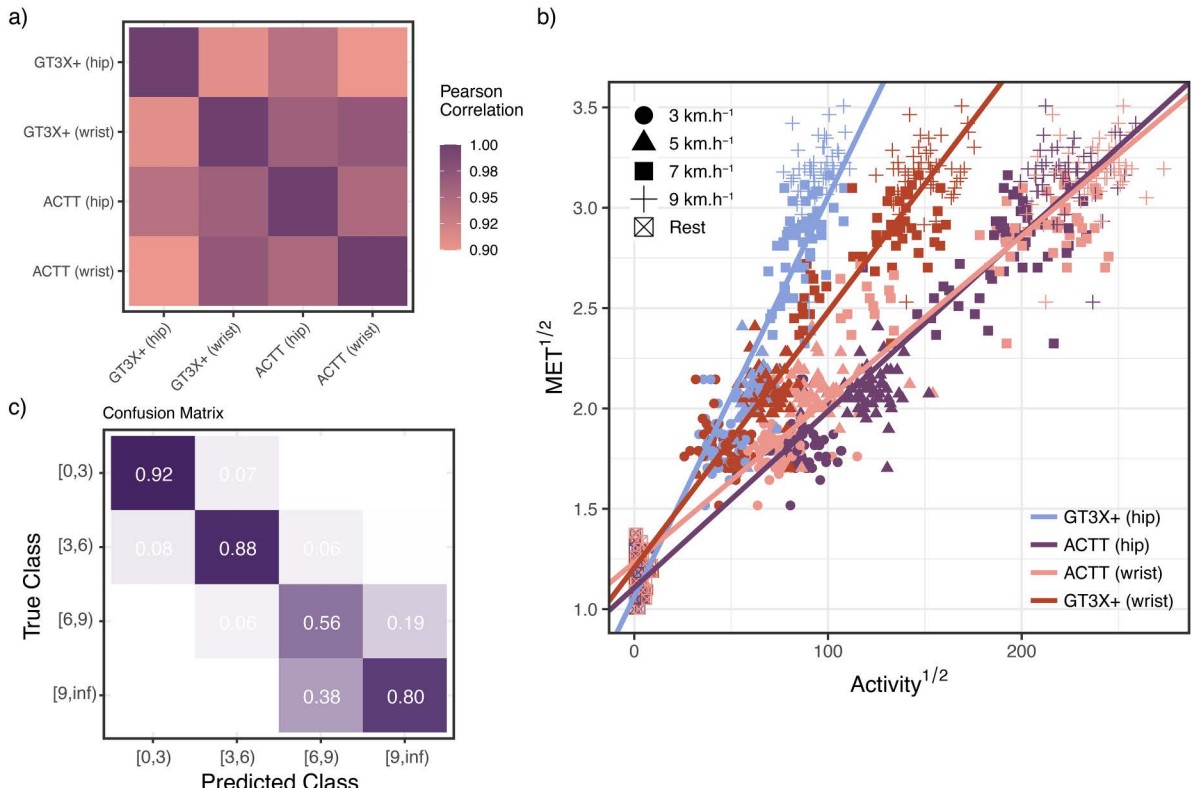

**Fig 2. Cross device score analyses. a)** Pearson correlation between the activity counts magnitudes measured at each device and placement combination. **b)** Model of Activity count and METs following the relationship Treadmill speed$^{1/2}$ ~ Activity$^{1/2}$ × Device$_{placement}$. **c)** Confusion matrix relating predicted intensity ranges from model in b) and metabolic equivalents ($VO^2$) obtained from simultaneous ergo-spirometry measurements, with a high true positive rate for the classes [0,3), [3,6) and [9,∞).

**Table 2. Cut-points and regression equations with 95% confidence intervals per device and placement.**

| | Cut-point, counts/min (95% CI) | | | MET$^{1/2}$ = $\beta_0$ + $\beta_1$ × Activity$^{1/2}$ | |
|---|---|---|---|---|---|
| Device | 3 METs | 6 METs | 9 METs | $\beta_0$ (95% CI) | $\beta_1$ (95% CI) |
| GT3X+ (hip) | 1132 (1041, 1223) | 4853 (4688, 5018) | 9468 (9135, 9800) | 1.062 (1.019, 1.105) | 0.0199 (0.0193, 0.0206) |
| ACTT (hip) | 5057 (4624, 5491) | 23339 (22522, 24157) | 46410 (44764, 48055) | 1.107 (1.066, 1.148) | 0.0088 (0.0085, 0.0091) |
| ACTT (wrist) | 3761 (3357, 4165) | 22368 (21503, 23232) | 47203 (45409, 48997) | 1.233 (1.195, 1.271) | 0.0081 (0.0079, 0.0084) |
| GT3X+ (wrist) | 1698 (1525, 1871) | 9503 (9143, 9862) | 19787 (19047, 20527) | 1.207 (1.168, 1.245) | 0.0127 (0.0123, 0.0132) |

**Table 3. Sensitivity, specificity, balanced accuracy, and area under the curve for the [0,3), [3,6), [6,9), and [9,∞) intensity ranges.**

| Metric | Class: [0,3) | Class: [3,6) | Class: [6,9) | Class: [9,∞) |
|---|---|---|---|---|
| Sensitivity | 0.8810 | 0.9190 | 0.6630 | 0.6441 |
| Specificity | 0.9762 | 0.9182 | 0.8932 | 0.9556 |
| Balanced Accuracy | 0.9286 | 0.9186 | 0.7781 | 0.7998 |
| AUC | 0.9820 | 0.6307 | 0.7813 | 0.9550 |

obtained confusion matrices show a consistent high accuracy for the [0,3); [3,6); and [9, ∞) classes, featuring true positive rates above 75%, both when aggregating all the devices and placements (see Fig 2b) and for specific combinations of device and placement (see Fig 3). Sensitivity and balanced accuracy are reduced for vigorous and very vigorous activity but remained well above 0.6, see Table 3.

## Discussion

This study compared the *ActTrust®* and *ActiGraph GT3X+®* activity counts during resting, treadmill walking and running. The main findings are as follows. Firstly, we established the suitability of our indirect calorimetry setup by measuring the metabolic equivalent as a function of the conditions (treadmill speed, resting). The speed of the treadmill was highly and significantly correlated with the estimated METs. Secondly, we demonstrated that movement counts increased as speed increased, and both *ActTrust®* and *GT3X+®* could differentiate among the different conditions, independently of the placement, either on the wrist or the hip. As a next step, we then modelled the association between METs and activity for each device X placement using a general linear model. In addition, we established cut points for movement counts of GT3X+ and ActTrust devices to classify the intensity of physical activity. These points were derived for the categories of moderate, intense and very intense physical activity (see Table 2), considering both the movement counts of the devices

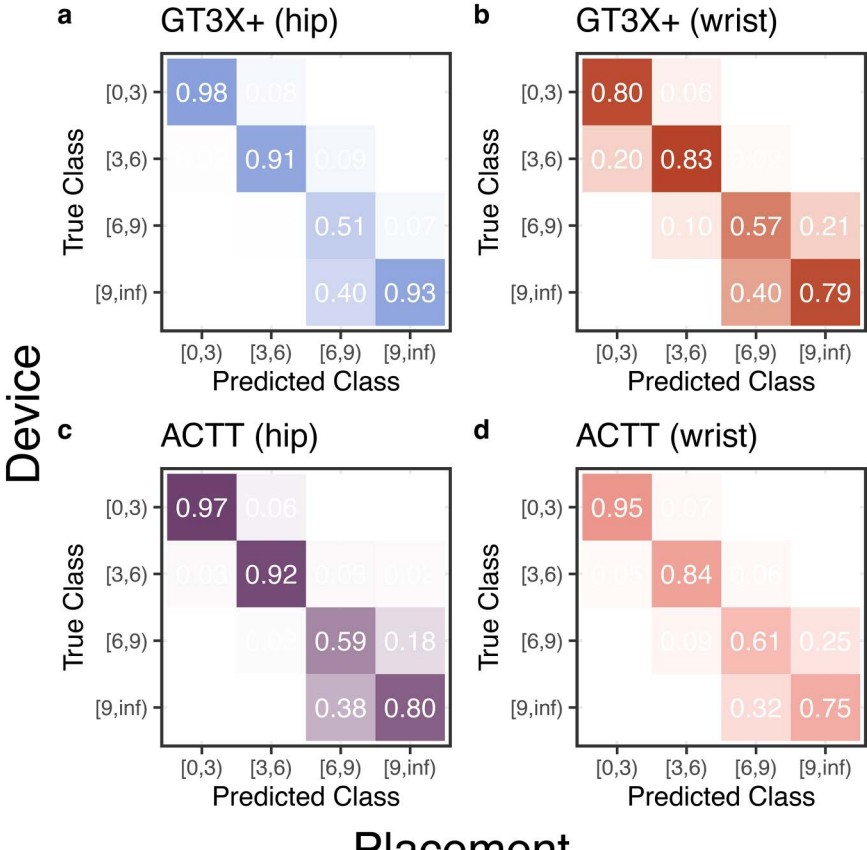

**Fig 3. Confusion matrices for each combination of device and placement, i.e., a) *GT3X+ (hip)*, b) *GT3X+ (wrist)*, c) *ACTT (hip)*, ACTT (wrist).** The individual devices confusion matrices reproduce a similar profile to the one observed for the entire model, with a higher true positive rate for the classes [0,3), [3,6) and [9,∞).

and the placement. Finally, we evaluated the ability of the new cut points from the linear regression models to accurately classify the intensity levels of physical activity objectively derived from EE measurement (METs). For this analysis, we calculated the balanced accuracy, sensitivity (recall), and specificity for each cut point and each linear model (device X placement).

Our results demonstrated that the movement count of both devices (GT3X+ and ActTrust), placed on both the hip and the wrist, increased with the increase in the velocity of the treadmill, as did the energy expenditure (METs) obtained from indirect calorimetry. Our results are consistent with similar validation studies [39,40]. If we compare our results with studies which used different devices [8,19], while our model's sensitivity and accuracy are reduced for vigorous and very vigorous activity, it still achieves accuracies above 0.90 and higher AUC scores than previous studies. Precisely, our model achieved area under the ROC curve of 0.98, 0.63, 0.78 and 0.95 for light, moderate, vigorous and very vigorous physical activity, respectively. As a comparison, the artificial neural network model developed by Santos-Lozano et al., resulted in AUC of 0.8, 0.6, 0.7 and 0.6 using a very similar protocol [8]. Therefore, we found that the model we proposed is accurate with a simpler method for the estimation of energy expenditure (EE) than the equations previously developed. Furthermore, our approach is fully explainable unlike neural network models.

The cut-points derived in the present study for the GT3X+ placed on the hip were lower than most published GT3X reference values across all thresholds. Compared to Sasaki et al. [19], our cut-points were 58%, 21%, and 2% lower at the MET 3, 6, and 9 thresholds, respectively, and 65%, 43%, and 18% lower than Santos-Lozano et al. [8] respective cut-off points for adults. Importantly, however, there is substantial variability in cut-points between these two quite similar studies, as they differ by 16% at MET 3, 39% at MET 6 and 20% at MET 9. The overall variability suggests that sample-level characteristics are the main drivers of the observed discrepancies rather than device or methodological artefacts.

Nevertheless, this is the first study to develop movement-based cut-off points for wrist and hip-worn ActTrust® devices. These cut-offs reflect moderate, intense and very intense physical activity categories. The study successfully classified physical activity intensity across the spectrum, demonstrating excellent accuracy. However, it is not possible to compare these values with other studies. For the discrimination between intensity categories based on METs, we propose thresholds to be used for the hip and wrist-worn ActTrust® devices. This allows researchers to examine how much walking people perform and distinguish it from running using these accelerometers.

Beyond estimating the intensity of physical activity, the value of validating ActTrust® lies in its ability to capture the entire 24-hour rest-activity cycle within a single device. Therefore, establishing PA cut-points for ActTrust® is a necessary first step toward leveraging its broader monitoring capabilities, which extend beyond exercise to the assessment of sleep and circadian rhythms. Actigraphy devices, such as the *ActTrust*® accelerometers have been used in sleep and circadian research for decades [41–43]. Actually, the sleep and wake states have been classically inferred by algorithms based on activity counts, provided frequency, duration and intensity of movement [44]. Although sleep studies are designed to collect 24-hour data, including rest and physical activity throughout the day, the simultaneous analysis of sleep/circadian data, physical activity, and sedentary behaviour is limited [13]. This gap is notable given strong evidence linking regular exercise, reduced sedentary behaviour, and healthy sleep to improved longevity and disease prevention [45,46].

Despite this clear dependence between physical activity and sleep, it is essential to move one step further and take advantage of the complete 24-hour rest-activity cycle data obtained through actigraphy. Precise health guidelines can only be determined by combining measurements of sleep, sedentary behaviour, and physical activity [13]. That said, expanding the availability of cut-off points for physical activity based on multiple actigraphy devices increases the potential to improve recommendations, guidelines and, ultimately, health and longevity. Therefore, studies like ours and the introduction of more affordable actigraphy devices, such as the ActTrust®, have the potential to stimulate future research by providing tools to extract physical activity and sleep information, even retrospectively. In fact, one of the main advantages of using accelerometry over self-reported (retrospective) physical activity is that it is not prone to recognition or memory biases, as recall measures are [47]. The validation of devices that were exclusively dedicated to inferring sleep-wake cycles as tools

to identify and classify physical activity according to the metabolic equivalent opens up the possibility of using such data not only prospectively, but also retrospectively. Studies such as ours are crucial to developing personalised programmes that maximise long-term adherence to physical activity and the long-term benefits of exercise. We acknowledge that adults should aim for at least 150–300 minutes of moderate-intensity aerobic physical activity, or 75–150 minutes of vigorous-intensity aerobic physical activity, or an equivalent combination throughout the week, to achieve substantial health benefits [48]. Likewise, adults should usually aim for 7 or more hours of regular sleep each night to promote optimal health [49]. However, only validated accelerometer devices can provide information on these two essential proxies of health simultaneously (sleep and physical activity).

It is important to acknowledge the limitations in the findings of this study. Although a significantly large number of participants (N = 56) was considered, the participants were young healthy adults, with no functional limitations, which limits generalisability. As a result, the findings may not be applicable to individuals with health conditions and other age groups without further validation. In addition, the study was carried out in a laboratory setting and focused on specific physical activities, walking and running, which may also limit the universality of our findings. Although these activities are common among populations around the world, it is crucial to assess physical activities in real life and home environments. Moreover, the fixed progressive protocol may have introduced order effects such as those linked to warm-up or fatigue. While this approach was chosen for safety, and mirrors clinical exercise testing protocols, future studies could employ counterbalanced designs or include longer rest periods to eliminate carryover effects.

Further work is needed to define appropriate cut-off points for other groups, such as children and older people, as well as different device placements, as our findings are limited to devices worn on the wrist and hip. Furthermore, the exercise protocol followed a fixed incremental order, which precludes ruling out potential order effects such as progressive fatigue or warm-up adaptation; future studies should consider counterbalanced or randomised condition sequences to address this limitation.

## Conclusion

In this study we have for the first time validated measures of the ActTrust®. When paired with a simple linear model, we can estimate energy expenditure and classify the intensity of physical activity. Our model distinguished between light, moderate, vigorous, and very vigorous activity intensity ranges with benchmark metrics comparable to or exceeding those reported in the literature, whilst adopting a much simpler and explainable model than black-box approaches such as neural networks, facilitating interpretation and implementation by practitioners. Nevertheless, we did not directly compare performance against such alternative modelling approaches in this study.

These results demonstrated the feasibility of using ActTrust® as a cost effective and scalable tool to estimate PA. It also has potential for potential for clinical use following validation in patient populations with diverse health conditions and functional capacities. Our results also provide hints on how to harmonise outcomes from different devices whilst estimating PA and analysing activity measures. Future work should extend our analyses to free-living conditions, comparison with commercial wearable devices other than GT3X+® and more naturalistic activity patterns.

## Supporting information

**S1 Table. ANOVA analysis and post-hoc Tukey's Honestly Significant Difference 316 (HSD) table.**
(CSV)

**S2 Table. Area under the curve (AUC) for one-vs-rest physical activity intensity classification stratified by sex.**
AUC values were computed for each intensity class using the model prediction score against the true class derived from indirect calorimetry for male (n = 34) and female (n = 22) participants. Intensity classes are defined as light [0,3), moderate [3,6), vigorous [6,9), and very vigorous [9,∞) METs.
(CSV)

**S1 Figure. Confusion matrices for physical activity intensity classification stratified by sex.** Row-normalised confusion matrices showing the proportion of observations assigned to each predicted intensity class relative to the true class derived from indirect calorimetry for (A) male (n = 34) and (B) female (n = 22) participants. Intensity classes are defined as light [0,3), moderate [3,6), vigorous [6,9), and very vigorous [9,∞) METs.
(PDF)

**S2 Figure. Bland-Altman plots comparing model-predicted and measured metabolic equivalents (METs) for each device and placement combination.** Each panel plots the difference between predicted and measured METs against their mean for each device and placement with GT3X + hip (A), GT3X+ wrist (B), ActTrust hip (C), and ActTrust wrist (D). Red dashed lines indicate the mean bias and blue dotted lines indicate the 95% limits of agreement. There is no evidence of systematic proportional bias across the range of measured values.
(PDF)

**S3 Figure. Regression diagnostics for the general model.** A) Normal Q-Q plot of model residuals with points following the reference blue line, featuring minor deviations at the tails. B) Residuals versus fitted values with no systematic trend supporting the assumption of homoscedasticity.
(PDF)

## Author contributions

**Conceptualization:** Elias dos Santos Batista, Mario Andre Leocadio-Miguel.

**Formal analysis:** Lucas G. S. França, Mario Andre Leocadio-Miguel.

**Investigation:** Elias dos Santos Batista, Ayrton Bruno de Morais Ferreira.

**Methodology:** Elias dos Santos Batista, Ayrton Bruno de Morais Ferreira, Lucas G. S. França, Mario Andre Leocadio-Miguel.

**Project administration:** Mario Andre Leocadio-Miguel.

**Supervision:** John Fontenele Araújo, Arnaldo Luis Mortatti, Mario Andre Leocadio-Miguel.

**Writing – original draft:** Elias dos Santos Batista, Stephania Ruth Basilio Silva Gomes, Ayrton Bruno de Morais Ferreira, Lucas G. S. França, John Fontenele Araújo, Arnaldo Luis Mortatti, Mario Andre Leocadio-Miguel.

**Writing – review & editing:** Lucas G. S. França, John Fontenele Araújo, Arnaldo Luis Mortatti, Mario Andre Leocadio-Miguel.

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
