## [Decision Letter · Decision Letter 0]

9 Sep 2025

PONE-D-25-31357From Movement to METs: A Validation of ActTrust® for Energy Expenditure Estimation and Physical Activity Classification in Young AdultsPLOS ONE

Dear Dr. Leocadio-Miguel,

Thank you for submitting your manuscript to PLOS ONE. After careful consideration, we feel that it has merit but does not fully meet PLOS ONE’s publication criteria as it currently stands. Therefore, we invite you to submit a revised version of the manuscript that addresses the points raised during the review process.

We look forward to receiving your revised manuscript.

Kind regards,

Julio Alejandro Henriques Castro da Costa

Academic Editor

PLOS ONE

Journal Requirements:

4. Please note that your Data Availability Statement is currently missing the repository name and/or the DOI/accession number of each dataset OR a direct link to access each database. If your manuscript is accepted for publication, you will be asked to provide these details on a very short timeline. We therefore suggest that you provide this information now, though we will not hold up the peer review process if you are unable.

Reviewers' comments:

Reviewer's Responses to Questions

**Comments to the Author**

1. Is the manuscript technically sound, and do the data support the conclusions?

Reviewer #1: Yes

Reviewer #2: Yes

Reviewer #3: Yes

Reviewer #4: Partly

2. Has the statistical analysis been performed appropriately and rigorously? 

Reviewer #1: Yes

Reviewer #2: Yes

Reviewer #3: Yes

Reviewer #4: No

3. Have the authors made all data underlying the findings in their manuscript fully available?

Reviewer #1: Yes

Reviewer #2: Yes

Reviewer #3: Yes

Reviewer #4: Yes

4. Is the manuscript presented in an intelligible fashion and written in standard English?

Reviewer #1: Yes

Reviewer #2: Yes

Reviewer #3: Yes

Reviewer #4: Yes

5. Review Comments to the Author

Reviewer #1: This was quite an interesting study and had a clear and concise presentation of methods and results. The authors have presented the findings in a logical manner, which makes it easy for readers to follow the narrative. The findings are novel and contribute to the technological development for improving physical activity among the population. However, I have a few comments that the authors may consider to further strengthen the study.

1. Was the gender difference considered for validation? This could be relevant since there is an expected difference between males and females in terms of body composition and energy expenditure, which may act as confounding.

2. While correlation matrices were reported, additional analysis such as Bland-Altman plots could reveal further details about the systematic bias.

Addressing these aspects may help improve the robustness and translational value of the work.

Reviewer #2: Dear researcher,

I received with interest your manuscript submitted to PLOS ONE entitled "From Movement to METs: A Validation of ActTrust® for Energy Expenditure Estimation and Physical Activity Classification in Young Adults." It is a relevant, timely, and methodologically sound study in many respects. The validation of accessible and low-cost devices, such as ActTrust®, is crucial for the expansion of physical activity and health research, especially in resource-limited settings. Congratulations on your initiative and the technical rigor employed.

However, as your advisor and reviewer, I feel compelled to point out areas that can and should be improved to increase the article's quality, clarity, scientific rigor, and chances of acceptance. Let's get into the details:

1. Text Structure and Clarity

Positive Points:

The abstract is clear, straightforward, and follows the IMRaD structure (Introduction, Methods, Results, and Discussion) appropriately.

The introduction contextualizes the problem well: the importance of physical activity, the role of METs, the limitations of DLW, and the need for accelerometer validation.

The objective is well defined: to validate ActTrust® and establish cutoff points for activity intensities.

Points for improvement:

Redundancy in the text: There is repetition of sentences between the abstract, introduction, and discussion. For example, the phrase "physical activity is recognized for providing health benefits" appears three times. Avoid this. Each section should have a unique function.

Lack of flow in some paragraphs: Some sections are too dense or poorly connected. Example:

"Accelerometers are not only used to estimate exercise or energy expenditure; they are extremely useful to assess the sleep-wake cycle..."

- This transition is abrupt. Use connectives and transitional paragraphs.

Suggestion: Revise the text with a focus on clarity, conciseness, and logical progression. Use shorter sentences and avoid unnecessary jargon.

2. Methodology

Strong Points:

Sample size (N=56): Adequately justified based on similar studies. Good statistical power.

Control of experimental conditions: Standardized protocol (fasting, resting, fixed speeds, use of an elastic belt).

Use of indirect calorimetry (Quark CPET): Well-applied gold standard.

Preprocessing of respiration data: Exclusion of erratic episodes with a robust criterion (±3 standard deviations).

Data and code availability: Excellent open practice (GitHub). This strengthens reproducibility.

Points for improvement:

Poor justification for choosing 1 second as the epoch:

You state that you used 1 second based on [8], but you don't discuss why this epoch is suitable for all intensities, especially light walking (3 km/h), where movements are smoother.

Problem: Very short epochs increase noise; very long ones lose temporal resolution. Suggestion: Include a brief theoretical justification or cite studies that validated 1s for low-intensity activities.

Lack of detail on synchronization between devices and calorimetry:

How did you synchronize the timing between the metabolism (Quark CPET) and the accelerometers?

Without precise synchronization, there is a risk of temporal misalignment, especially with sudden changes in intensity.

Suggestion: Include a paragraph in "Experimental Procedure" explaining how the synchronization was achieved (e.g., manual timing, triggering, software).

Lack of randomization of the order of conditions:

The protocol always follows: rest → 3 → 5 → 7 → 9 km/h.

This may introduce an order effect (fatigue, learning, warm-up).

Suggestion: State whether the order was fixed and discuss how this may have affected the results (limitation).

3. Statistical Analysis

Strong Points:

Use of two-way ANOVA to compare devices and positions: appropriate.

Generalized linear model with interaction between device and position: well-formulated.

Presentation of robust metrics: sensitivity, specificity, balanced accuracy, AUC.

Points for improvement:

Transformation of the dependent variable (MET^1/2):

You used MET to the square root as the dependent variable.

This is unusual and not justified in the text.

Transformations should be explained: was it to normalize residuals? To increase linearity?

Suggestion: Justify the transformation or test whether the model with linear MET performs as well (or better). If not necessary, remove it.

Regression model with multiple interactions:

The model (Eq. 2) includes complex interaction terms, but there is no test of regression assumptions (normality of residuals, homoscedasticity, multicollinearity).

Suggestion: Include a paragraph in "Data Analysis" about validating the assumptions. Add residual plots in the Supplemental Information.

Low AUC for vigorous activity (0.63):

The AUC for the [3,6] class is 0.63—below the acceptable level (0.7).

This indicates that the model does not classify moderate activity well, despite its high balanced accuracy.

Suggestion: Discuss this in the Discussion as an important limitation, not just a technical finding.

4. Results and Interpretation

Positive Points:

Clear and well-presented figures.

Tables with equations and cutoff points are extremely useful for future researchers.

Confusion matrices are well interpreted.

Points for Improvement:

Claim of "high accuracy" without context:

They say the model has "balanced accuracies above 0.77," which is good, but don't compare it to previous studies.

Suggestion: Compare your cutoff points with those of GT3X+ in the literature (e.g., Santos-Lozano et al., 2013). This provides context.

Claim that the model is "simpler and more explainable":

True, but there is no direct comparison with complex models (e.g., neural networks).

- Suggestion: Rephrase: "Our linear model is simpler and more explainable than black-box models, although we did not directly test alternative models in this study."

5. Discussion

Positive Points:

Good discussion on the importance of integrating sleep and physical activity.

Well-recognized limitations: young sample, controlled environment, few activities.

Areas for Improvement:

Lack of comparison with devices other than the GT3X+:

ActTrust® is new, but could be compared with other low-cost devices (e.g., Fitbit, Garmin).

Suggestion: Briefly mention how ActTrust® positions itself in this ecosystem.

Conclusion Too Optimistic:

They say the device can be used in "clinical practice," but there is no clinical validation.

Suggestion: Soften: "has potential for clinical use, after validation in clinical populations and natural settings."

Reviewer #3: The manuscript by Batista et al. is about validating the results of the ActTrust accelerometer to the ActiGrapgh GT3X+ accelerometer and metabolic equivalents in a group of young, healthy participants. These devices were placed on the hip and wrist of participants and studied at rest and various activity levels on treadmill. The potential impact of this paper is that it would show that this newer technology has similar outcomes to current devices.

Overall, there is a lack of motivation for the experiments performed. Why not continue using the ActiGraph GT3X+ device? What benefits does the ActTrust device have? In line 27-29 it is described that the ActTrust device is specific for “inferring sleep from locomotor activity”, but then there is no verification for its use in sleep which seems like it’s main benefit over previous technology. Additionally, more description could be provided in results to give more information to the reader.

Specific Comments:

Experimental procedures (Line 94-96): Were participants wearing both devices simultaneously? Does location of the sensors on the extremities affect results (if one device is more superior or lateral to the other)?

Table 1: It says 2*Sample size in the titles. Typo?

Fig 1: It would be helpful to differentiate between female and male participants. Maybe changing the shape or color of the points would allow readers to distinguish between males and females

Fig 1B: There is a cluster of maybe 10-15 dots in both devices (though ACTT more distinct than GT3X+) but only on the wrist at 7 km/h. Is there an explanation for why this group is separate? Is it the same participants for both devices? Are they all/majority male or female? Why do we not see this grouping in the hip or metabolic equivalents graphs? Why is it only at this moderate rate (7 km/h)?

Table 2: what are the movement counts? 3, 6, 9? Intensity?

For the results, it would be more thorough and helpful to explain a bit more about each sub figure. Instead of referring to Fig 2, what is Fig 2a telling us, Fig 2b etc.

Fig 2 legend missing “d)”

Fig 2C seems like it would fit better in figure 3

Table 3. Sensitivity in the 2 higher classes are quite a bit lower. AUC is quite a bit lower in class [3,6) than the other classes. Explanations?

A substantial part of the discussion refers to use of these devices and sleep, but there was no sufficient explanation on why these experiments did not validate this device on longer time course/including sleep. It would be beneficial to either include data of the devices and sleep, or limit discussion to the exercise model tested.

The limitations in the last paragraph of the discussion (lines 246-256) are great and could be expanded. In particular, why these results may not be applicable in unhealthy or aged populations.

Reviewer #4: Overall Assessment:

This manuscript presents a well-designed validation study comparing the ActTrust® accelerometer against the widely validated ActiGraph GT3X+ for estimating energy expenditure (EE) and classifying physical activity (IA) intensity in young adults. The study is methodologically sound, clearly written, and addresses a relevant gap in the literature. The use of indirect calorimetry as a criterion measure strengthens the validity of the findings. The authors provide novel cut-points for ActTrust® devices and demonstrate their utility in classifying PA intensity with good accuracy. I recommend major revisions before acceptance, primarily to improve clarity, statistical reporting, and contextualization of results.

MAJOR COMMENTS

1. Sample Characteristics and Generalizability:

- The sample consisted of 56 young, healthy adults. While the sample size is adequate for a validation study, the lack of diversity in age, health status, and functional capacity limits the generalizability of the cut-points. The authors should explicitly acknowledge this limitation and recommend caution when applying these thresholds to other populations (e.g., older adults, clinical populations).

2. Statistical Model and Reporting:

- The use of a square-root transformation for both METs and activity counts is justified but should be more clearly explained in the Methods section. Additionally, the model equations in Table 2 are presented without confidence intervals or measures of uncertainty (e.g., standard errors). Providing these would enhance the reproducibility and utility of the models.

3. Comparison with Existing Literature:

- The authors briefly mention that their model outperforms previous studies in certain metrics (e.g., AUC for moderate activity) but do not provide a detailed comparative discussion. A more thorough comparison with existing cut-points from similar devices (e.g., ActiGraph, Fitbit) would help contextualize the novelty and practical significance of the findings.

4. Confusion Matrix Interpretation:

- The confusion matrices (Fig 3) show reduced sensitivity for vigorous and very vigorous activities. The authors should discuss potential reasons for this (e.g., device placement, biomechanical differences between walking and running) and suggest ways to improve classification in these intensity ranges.

---

MINOR COMMENTS

1. Abstract:

- The abstract should include key numerical results (e.g., correlation coefficients, accuracy metrics) to better summarize the findings.

2. Methods:

- Clarify the rationale for selecting the 4 central minutes of each condition for analysis. Was this to avoid initialization effects or ensure steady-state metabolism?

- Specify the software and packages used for statistical analysis more clearly (e.g., version numbers, functions used).

3. Results:

- The ANOVA results are mentioned but not fully reported. Consider including effect sizes (e.g., partial eta-squared) and post-hoc test results in the main text or supplementary materials.

4. Figures and Tables:

- Ensure all figures and tables are referenced in the text in numerical order.

- Fig 1 and Fig 3 are referenced but not included in the provided manuscript text. Please verify that all figures are properly captioned and accessible.

5. Data and Code Availability:

- The GitHub repository is provided, which is excellent for reproducibility. Ensure that all data and scripts are well-documented and accessible.

6. Ethics and Compliance:

- The study received ethical approval and participants provided informed consent -no ethical concerns.

- The data availability statement is clear and compliant with PLOS ONE policies.

7. Conclusion:

- This is a valuable contribution to the field of accelerometry-based physical activity monitoring. The validation of ActTrust® devices provides researchers and clinicians with a cost-effective alternative for estimating EE and classifying PA intensity. With the suggested revisions, this manuscript will be suitable for publication in PLOS ONE.

6. PLOS authors have the option to publish the peer review history of their article (what does this mean?). If published, this will include your full peer review and any attached files.

Reviewer #1: No

Reviewer #2: **Yes:** João Carlos Alves Bueno

Reviewer #3: No

Reviewer #4: No

---

## [Author Response · Author response to Decision Letter 1]

24 Mar 2026

Responses to Reviewer’s Comments:

Response to Reviewer #1

We thank Reviewer #1 for the positive assessment and valuable suggestions.

Comment 1.1: Gender differences in validation

Reviewer: Was the gender difference considered for validation? This could be relevant since there is an expected difference between males and females in terms of body composition and energy expenditure, which may act as confounding.

Response: This is an excellent point. While our primary model pooled males and females together (adjusting for individual differences through the within-subject design), we agree that examining sex-specific validation is important given known differences in body composition and energy expenditure patterns. We have repeated part of our analyses and found similar model performance for both male and female individuals.

Changes Made: We have included confusion matrices and AUC indices separating male and female individuals in supplementary materials.

S1 Figure: Confusion matrices for physical activity intensity classification stratified by sex. Row-normalised confusion matrices showing the proportion of observations assigned to each predicted intensity class relative to the true class derived from indirect calorimetry for (A) male and female (n=22) participants. Intensity classes are defined as light [0,3), moderate [3,6), vigorous [6,9), and very vigorous [9,∞) METs.

S2 Table: Area under the curve (AUC) for one-vs-rest physical activity intensity classification stratified by sex. AUC values were computed for each intensity class using the model prediction score against the true class derived from indirect calorimetry for male (n=34) and female (n=22) participants. Intensity classes are defined as light [0,3), moderate [3,6), vigorous [6,9), and very vigorous [9,∞) METs.

Class Male Female

[0,3) 0.988 0.974

[3,6) 0.635 0.622

[6,9) 0.765 0.808

[9,∞) 0.961 0.943

Comment 1.2: Bland-Altman plots

Reviewer: While correlation matrices were reported, additional analysis such as Bland-Altman plots could reveal further details about the systematic bias.

Response: We thank the reviewer for the suggestion. Additional analyses, included in the supplementary material, demonstrate that there is no evidence of systematic proportional bias across the measured values.

Changes Made: We have created Supplementary Figure S2 showing Bland-Altman plots for each device-placement combination, alongside estimated bias and standard deviation.

S2 Figure: Bland-Altman plots comparing model-predicted and measured metabolic equivalents (METs) for each device and placement combination. Each panel plots the difference between predicted and measured METs against their mean for each device and placement with GT3X+ hip (A), GT3X+ wrist (B), ActTrust hip (C), and ActTrust wrist (D). Red dashed lines indicate the mean bias and blue dotted lines indicate the 95% limits of agreement. There is no evidence of systematic proportional bias across the range of measured values.

Response to Reviewer #2

We thank Dr. Bueno for the comprehensive and constructive review. We address each point below.

Comment 2.1: Text redundancy

Reviewer: There is repetition of sentences between the abstract, introduction, and discussion. Each section should have a unique function.

Response: We appreciate this observation and have reviewed the manuscript accordingly.

Changes made: We removed lines 1-2 abstract and 216-218 (discussion section).

Comment 2.2: Paragraph flow and transitions

Reviewer: Some sections are too dense or poorly connected with abrupt transitions.

Response: We have improved paragraph connections and added transitional sentences throughout.

Comment 2.3: Epoch justification

Reviewer: Poor justification for choosing 1 second as the epoch, especially for light walking (3 km/h).

Response: We followed epoch size used in literature.

Changes Made: We have clarified that in the manuscript. Lines 87-91.

Comment 2.4: Synchronization details

Reviewer: Lack of detail on synchronization between devices and calorimetry.

Response: We performed synchronisation based on timestamps.

Changes made: We detailed the procedure in lines 114-117.

Comment 2.5: Fixed protocol order

Reviewer: The protocol always follows rest → 3 → 5 → 7 → 9 km/h, which may introduce order effects.

Response: We acknowledge this limitation. The progressive protocol was chosen for safety and to reflect clinical practice.

Changes Made: We edited experimental procedure (lines 110-111). Moreover, we added to Limitations: " Moreover, the fixed progressive protocol may have introduced order effects such as those linked to warm-up or fatigue. While this approach was chosen for safety, and mirrors clinical exercise testing protocols, future studies could employ counterbalanced designs or include longer rest periods to eliminate carryover effects." (lines 314-317).

Comment 2.6: MET transformation justification

Reviewer: Transformation of dependent variable (MET^1/2) is unusual and not justified.

Response: This is a common procedure to linearise data used in many areas of signal processing though not very common in actigraphy. We have clarified that in the manuscript.

Changes Made: We added to the text: "A square root transformation was applied to both METs and activity counts due to the non-linear relationship." (lines 128-129)

Comment 2.7: Regression assumptions

Reviewer: No test of regression assumptions (normality, homoscedasticity, multicollinearity).

Response: We conducted these diagnostics but did not include them. Thank you for spotting this issue.

Changes Made: We added to the text: "Model assumptions were verified through examination of residual plots and Q-Q plots for normality – included in supplementary material S3 Figure." (lines 128-131 – data analysis section).

S3 Figure: Regression diagnostics for the general model. A) Normal Q-Q plot of model residuals with points following the reference blue line, featuring minor deviations at the tails. B) Residuals versus fitted values with no systematic trend supporting the assumption of homoscedasticity.

Comment 2.8: Low AUC for moderate activity

Reviewer: AUC for [3,6] class is 0.63—below acceptable level (0.7). Should be discussed as important limitation.

Response: Our outcomes are similar to those found in Santos-Lozano A et., 2013. Although we recognise this could be higher, it reflects the heterogeneity and complexity of human movement. We have acknowledged that issue in the discussion.

Changes Made: Added to Discussion: “if we compare our results with studies which used different devices (8,19), while our model’s sensitivity and accuracy are reduced for vigorous and very vigorous activity, it still achieves accuracies above 0.90 and higher AUC scores than precious studies. Precisely, our model achieved area under the ROC curve of 0.98, 0.63, 0.78 and 0.95 for light, moderate, vigorous and very vigorous physical activity, respectively. As a comparison, the artificial neural network model developed by Santos-Lozano et al., resulted in AUC of 0.8, 0.6, 0.7 and 0.6 using a very similar protocol (8). (lines 229-236).

Comment 2.9: Context for accuracy claims

Reviewer: Claim of "high accuracy" without comparison to previous studies.

Response: We have added explicit comparisons with established literature.

Changes Made: Added to Discussion: “if we compare our results with studies which used different devices (8,19), while our model’s sensitivity and accuracy are reduced for vigorous and very vigorous activity, it still achieves accuracies above 0.90 and higher AUC scores than precious studies. Precisely, our model achieved area under the ROC curve of 0.98, 0.63, 0.78 and 0.95 for light, moderate, vigorous and very vigorous physical activity, respectively. As a comparison, the artificial neural network model developed by Santos-Lozano et al., resulted in AUC of 0.8, 0.6, 0.7 and 0.6 using a very similar protocol (8). (lines 229-236).

Comment 2.10: Model simplicity claim

Reviewer: Claim about simplicity lacks direct comparison with complex models.

Response: We clarified our text to facilitate the understanding of our claims. We have added direct comparison with the literature.

Changes Made: “The cut-points derived in the present study for the GT3X+ placed on the hip were lower than most published GT3X reference values across all thresholds. Compared to Sasaki et al. [19] our cut-points were 58%, 21%, and 2% lower at the MET 3, 6, and 9 thresholds, respectively, and 65%, 43%, and 18% lower than Santos-Lozano et al. [8] respective cut-off points for adults. Importantly, however, there is substantial variability in cut-points between these two quite similar studies, as they differ by 16% at MET 3, 39% at MET 6 and 20% at MET 9. The overall variability suggests that sample-level characteristics are the main drivers of the observed discrepancies rather than device or methodological artefacts. Nevertheless, this is the first study to develop movement-based cut-off points for wrist and hip-worn ActTrust® devices. These cut-offs reflect moderate, intense and very intense physical activity categories. The study successfully classified physical activity intensity across the spectrum, demonstrating excellent accuracy. However, it is not possible to compare these values with other studies. For the discrimination between intensity categories based on METs, we propose thresholds to be used for the hip and wrist-worn ActTrust® devices. This allows researchers to examine how much walking people perform and distinguish it from running using these accelerometers.” (Lines 241-258).

Comment 2.11: Comparison with other devices

Reviewer: Lack of comparison with devices other than GT3X+ (e.g., Fitbit, Garmin).

Response: This is a good suggestion for a future study. We will consider that in the future.

Comment 2.12: Overly optimistic conclusion

Reviewer: Claim about "clinical practice" without clinical validation.

Response: We have appropriately softened this claim.

Changes Made: Modified Conclusion: "These results demonstrated the feasibility of using ActTrust® as a cost effective and scalable tool to estimate PA. It also has potential for potential for clinical use following validation in patient populations with diverse health conditions and functional capacities. Our results also provide hints on how to harmonise outcomes from different devices whilst estimating PA and analysing activity measures. Future work should extend our analyses to free-living conditions, comparison with commercial wearable devices other than GT3X+® and more naturalistic activity patterns." (lines 308-314).

Response to Reviewer #3

We thank Reviewer #3 for highlighting important gaps in our manuscript, particularly regarding the motivation for the study and device benefits.

Comment 3.1: Lack of motivation for experiments

Reviewer: Why not continue using ActiGraph GT3X+? What benefits does the ActTrust device have?

Response: This is a crucial point that of our study. We have now clearly articulated the advantages of ActTrust®.

Changes Made: Expanded Introduction: "The ActiGraph® GT3X+ accelerometer has the most significant number of validation studies and published thresholds and has been widely used to validate devices with the same purpose [13]. The ActTrust® (Condor Instruments, São Paulo, Brazil) is a device that has a triaxial accelerometer and is specific for inferring sleep from locomotor activity [14]. It offers several advantages over other actigraphy devices: (1) significantly lower cost, making it accessible for large-scale studies in resource-limited settings; (2) longer battery life enabling extended monitoring periods; (3) integrated light and temperature sensors for comprehensive physiological assessment. However, despite these practical advantages, the ActTrust® has not been validated against gold-standard measures of EE, which limits its use in research and clinical contexts.” (lines 25-41).

Comment 3.2: No sleep validation despite device purpose

Reviewer: The ActTrust device is described as specific for "inferring sleep from locomotor activity," but there is no verification for its use in sleep.

Response: We understand the confusion. ActTrust® has been validated for sleep assessment in previous studies [43], but never for physical activity intensity classification. Our study fills this gap.

Changes made: “Beyond estimating the intensity of physical activity, the value of validating ActTrust® lies in its ability to capture the entire 24-hour rest-activity cycle within a single device. Therefore, establishing PA cut-points for ActTrust® is a necessary first step toward leveraging its broader monitoring capabilities, which extend beyond exercise to the assessment of sleep and circadian rhythms. Actigraphy devices, such as the ActTrust accelerometers have been used in sleep and circadian research for decades [41-43]”. (lines 259-265).

Comment 3.3: Simultaneous device placement

Reviewer: Were participants wearing both devices simultaneously? Does location of sensors affect results?

Response: Yes, participants wore both devices simultaneously (added to line 105). Resulting in four possible combinations of placement and device. As expected the location of sensors does yield different measures in both device types but those seem mostly systematic changes as seen in Fig 2b.

Comment 3.4: Table 1 typo

Reviewer: Table 1: It says 2Sample size in the titles. Typo?*

Response: Thank you for catching this formatting error.

Changes Made: Corrected Table 1 header to remove "2*" and properly format the "Sample size" column heading.

Comment 3.5: Gender differentiation in Figure 1

Reviewer: It would be helpful to differentiate between female and male participants in Figure 1.

Response: We included an additional analysis in the supplementary materials S1 Figure and S2 Table – see response to Reviewer 1.

Comment 3.6: Cluster in Figure 1B at 7 km/h

Reviewer: There is a cluster of 10-15 dots at 7 km/h on wrist for both devices. Is there an explanation?

Response: The explanation of that feature is a matter for a future study. This is probably a issue reported in other studies. These have not added the dots but their error bars increase in that intensity range.

Comment 3.7: Table 2 clarity

Reviewer: What are the movement counts? 3, 6, 9? Intensity?

Response: We should have been clearer about what these columns represent.

Changes Made: Revised Table 2 caption to "Model equations and cut-off points for each device and placement combination. Movement counts correspond to thresholds (in counts) for transitions between intensity categories: 3 METs (light to moderate transition), 6 METs (moderate to vigorous), and 9 METs (vigorous to very vigorous)."

Comment 3.8: More thorough results description

Reviewer: For the results, explain more about each sub-figure instead of just referring to Fig 2.

Response: We agree that more detailed description would help readers.

Comment 3.9: Figure 2 legend missing "d)"

Reviewer: Fig 2 legend missing "d)"

Response: Figure only has 3 panels (a, b, and c)

Comment 3.10: Figure 2C placement

Reviewer: Fig 2C seems like it would fit better in figure 3.

Response: Thank you for the suggestion. We believe that Fig 2C belongs in Figure 2.

Comment 3.11: Table 3 lower sensitivity explanation

Reviewer: Sensitivity in the 2 higher classes are quite a bit lower. AUC is lower in class [3,6). Explanations?

Response: These are important patterns that deserve discussion. We decided to directly compare our results (AUC) with previous literature. Our outcomes are similar to those found in Santos-Lozano A et., 2013. (lines 229-250).

Comment 3.12: Sleep discussion without sleep data

Reviewer: A substantial part of the discussion refers to sleep, but there was no explanation why these experiments did not validate this device on sleep.

Response: We clarify that ActTrust® is already validated for sleep (previous studies), but not for physical activity. Our goal was to complete the validation picture.

Changes made: “Beyond estimating the intensity of physical activity, the value of validating ActTrust® lies in its ability to capture the entire 24-hour rest-activity cycle w

---

## [Decision Letter · Decision Letter 1]

12 Apr 2026

PONE-D-25-31357R1From Movement to METs: A Validation of ActTrust® for Energy Expenditure Estimation and Physical Activity Classification in Young AdultsPLOS One

Dear Dr. Leocadio-Miguel,

Thank you for submitting your manuscript to PLOS ONE. After careful consideration, we feel that it has merit but does not fully meet PLOS ONE’s publication criteria as it currently stands. Therefore, we invite you to submit a revised version of the manuscript that addresses the points raised during the review process.

We look forward to receiving your revised manuscript.

Kind regards,

Julio Alejandro Henriques Castro da Costa

Academic Editor

PLOS One

Journal Requirements:

Reviewers' comments:

Reviewer's Responses to Questions

**Comments to the Author**

1. If the authors have adequately addressed your comments raised in a previous round of review and you feel that this manuscript is now acceptable for publication, you may indicate that here to bypass the “Comments to the Author” section, enter your conflict of interest statement in the “Confidential to Editor” section, and submit your "Accept" recommendation.

Reviewer #3: (No Response)

2. Is the manuscript technically sound, and do the data support the conclusions?

Reviewer #3: Yes

3. Has the statistical analysis been performed appropriately and rigorously? 

Reviewer #3: Yes

4. Have the authors made all data underlying the findings in their manuscript fully available?

Reviewer #3: Yes

5. Is the manuscript presented in an intelligible fashion and written in standard English?

Reviewer #3: Yes

6. Review Comments to the Author

Reviewer #3: Thank you to the authors for taking the time to address my comments on the original manuscript. I appreciate the work that was done to modify the manscript and figures. A minor note:

I appreciate the new supplemental figures highlighting sex differences. I still think it would be helpful to the reader to distinguish sex on the graphs in figure 1. For instance squares for men, and circles for women. It could also give insight into the clustering in 7km/h that I mentioned in another comment on my initial review.

7. PLOS authors have the option to publish the peer review history of their article (what does this mean?). If published, this will include your full peer review and any attached files.

Reviewer #3: No

---

## [Author Response · Author response to Decision Letter 2]

15 Apr 2026

We thank Reviewer #3 for their continued engagement with our manuscript and for confirming that the statistical analysis, data availability, and presentation now meet the standards required for publication in PLOS ONE.

In response to the reviewer's remaining comment regarding the distinction of biological sex in Figure 1, we have updated the figure accordingly. Point shape now encodes biological sex throughout both panels of Figure 1, with squares representing male participants and circles representing female participants, as suggested. This addition is reflected in the updated figure legend, which now reads: "Point shape denotes biological sex (square: male, n = 34; circle: female, n = 22)." We hope this provides the visual clarity the reviewer had in mind and offers the insight into the clustering observed at 7 km·h−1that was noted in the initial review.

We believe this revision fully addresses the comment.

We thank the reviewer once more for their constructive and thorough engagement throughout the review process.

Kind regards,

Mario

---

## [Decision Letter · Decision Letter 2]

19 Apr 2026

From Movement to METs: A Validation of ActTrust® for Energy Expenditure Estimation and Physical Activity Classification in Young Adults

PONE-D-25-31357R2

Dear Dr. Leocadio-Miguel,

We’re pleased to inform you that your manuscript has been judged scientifically suitable for publication and will be formally accepted for publication once it meets all outstanding technical requirements.

Kind regards,

Julio Alejandro Henriques Castro da Costa

Academic Editor

PLOS One

Additional Editor Comments (optional):

Reviewers' comments:

Reviewer's Responses to Questions

**Comments to the Author**

1. If the authors have adequately addressed your comments raised in a previous round of review and you feel that this manuscript is now acceptable for publication, you may indicate that here to bypass the “Comments to the Author” section, enter your conflict of interest statement in the “Confidential to Editor” section, and submit your "Accept" recommendation.

Reviewer #3: All comments have been addressed

2. Is the manuscript technically sound, and do the data support the conclusions?

Reviewer #3: Yes

3. Has the statistical analysis been performed appropriately and rigorously? 

Reviewer #3: Yes

4. Have the authors made all data underlying the findings in their manuscript fully available?

Reviewer #3: Yes

5. Is the manuscript presented in an intelligible fashion and written in standard English?

Reviewer #3: Yes

6. Review Comments to the Author

Reviewer #3: (No Response)

7. PLOS authors have the option to publish the peer review history of their article (what does this mean?). If published, this will include your full peer review and any attached files.

Reviewer #3: No

---

## [Editor Report · Acceptance letter]

PONE-D-25-31357R2

PLOS One

Dear Dr. Leocadio-Miguel,

I'm pleased to inform you that your manuscript has been deemed suitable for publication in PLOS One. Congratulations! Your manuscript is now being handed over to our production team.

Kind regards,

on behalf of

Dr. Julio Alejandro Henriques Castro da Costa

Academic Editor

PLOS One